# The Role of Extracellular Vesicles in Mediating Signaling in Biliary Epithelial Cell Activation and Cholangiopathies

**DOI:** 10.3390/cells14161274

**Published:** 2025-08-18

**Authors:** Sharmila Fagoonee, Marcela Fabiana Bolontrade, Paola Defilippi, Ralf Weiskirchen

**Affiliations:** 1Institute of Biostructure and Bioimaging (CNR) and Molecular Biotechnology Center “Guido Tarone”, 10126 Turin, Italy; 2Institute of Translational Medicine and Biomedical Engineering (IMTIB)—CONICET, Buenos Aires C1199ACL, Argentina; marcela.bolontrade@hospitalitaliano.org.ar; 3University of the Italian Hospital (UHI)—Italian Hospital of Buenos Aires (HIBA), Buenos Aires C1199ACL, Argentina; 4Molecular Biotechnology Center “Guido Tarone”, Department of Molecular Biotechnology and Health Sciences, University of Turin, 10126 Turin, Italy; paola.defilippi@unito.it; 5Institute of Molecular Pathobiochemistry, Experimental Gene Therapy and Clinical Chemistry (IFMPEGKC), RWTH University Hospital Aachen, D-52074 Aachen, Germany

**Keywords:** biliary tract diseases, secretome, cholangiocyte, extracellular vesicles

## Abstract

Cholangiopathies, a diverse group of diseases affecting the biliary tract, are characterized by the activation of cholangiocytes, fibrosis, and inflammation. Recent research has identified extracellular vesicles (EVs) as crucial mediators of communication within the hepatobiliary system. This review aims to explore the impact of EVs on cholangiocyte behavior and their role in disease development. EVs originating from cholangiocytes, hepatocytes, and immune cells carry a variety of molecules, including non-coding RNAs, proteins, and lipids, which influence immune responses, fibrosis, and epithelial repair. Specifically, EVs released by activated or senescent cholangiocytes can worsen inflammation and fibrosis by delivering molecules such as lncRNA H19, miR-21, and damage-associated molecular patterns (DAMPs) to hepatic stellate and immune cells. Additionally, the polarity and content of EVs are influenced by specific subcellular domains of cholangiocytes, indicating distinct signaling functions. In conditions such as primary sclerosing cholangitis (PSC), cholangiocarcinoma (CCA), and biliary atresia, EVs play a role in disease progression and offer potential as non-invasive biomarkers and therapeutic targets. This review underscores the importance of in-depth profiling and validation of EVs to fully utilize their diagnostic and therapeutic capabilities. Overall, EV-mediated signaling is a critical mechanism in cholangiopathies, providing a new avenue for understanding disease progression and developing precision medicine approaches.

## 1. Introduction

Biliary tract diseases, or cholangiopathies, result from chronic conditions that affect the epithelial cells lining the intrahepatic and extrahepatic bile ducts (cholangiocytes). They encompass a heterogeneous group of biliary disorders that can be broadly classified into genetic, idiopathic, malignant, and miscellaneous etiologies [1]. In the pediatric population, bile flow may be impeded by immune, genetic, or infectious causes, leading to conditions such as biliary atresia. Primary biliary cholangitis (PBC) is encountered in adults, while primary sclerosing cholangitis (PSC) can be found in both populations. Cholangiocarcinoma (CCA) is one of the deadliest cancers, still awaiting early diagnostic tools and improvements in therapeutic strategies. Cholangiopathies exhibit a convergent pathophysiological profile marked by cholestasis of varying severity, progressive loss of intrahepatic bile ducts (ductopenia), compensatory bile duct proliferation (bile duct hyperplasia), and sustained inflammatory activity within the portal and periportal regions [2]. Over time, these processes promote fibrogenesis, which may culminate in biliary cirrhosis and ultimately progress to end-stage liver disease, requiring liver transplantation as the ultimate therapeutic intervention (Figure 1).

Cholangiopathies account for the majority of pediatric liver transplants, comprising approximately 80% of cases, and contribute significantly to adult liver transplant indications, representing around 10–20% of procedures [3]. However, liver transplantation may cause biliary complications in 5–35% of recipients due to bile leakage and anastomotic and non-anastomotic biliary strictures [4]. Several chronic hepatic diseases, including metabolic conditions such as metabolic dysfunction-associated fatty liver disease (MAFLD, previously known as NAFLD) and its progressive form, metabolic associated steatohepatitis (MASH, previously known as MASH), as well as alcohol-associated liver disease, are also subject to cholangiocyte activation-related pathogenesis. In fact, the development of portal fibrosis has been associated with the advancement of MAFLD to MASH [5]. The management of cholangiopathies remains challenging due to their heterogeneous pathogenesis and limited effective medical therapies. Current strategies aim to reduce cholestasis, modulate immune responses, and prevent fibrogenic progression, while emerging therapies are increasingly focused on host–microbiota interactions, which are now recognized as contributors to biliary pathophysiology [6,7,8,9].

Thus, cholangiocytes, a small population of liver cells, appear to play a crucial role in the progression of biliary pathogenesis. This highlights the need for increased focus on exploring new therapeutic strategies and biomarkers. Cholangiocytes can become activated not only due to alterations in signaling pathways but also in response to factors present in the secretome, which can further worsen hepatobiliary diseases. The purpose of this concise review is to provide an overview of how the extracellular vesicles (EVs) component of the secretome can impact cholangiocyte activation and contribute to hepatobiliary diseases.

## 2. Cholangiocyte Function and Activation

Cholangiocytes are a highly heterogeneous population of biliary epithelial cells that vary in structure and metabolic activities. Large cholangiocytes are mainly found along the proximal, larger-caliber bile ducts and are known for their well-developed organelles, abundant cytoplasm, and complex apical specializations, indicating their active involvement in bile modification and electrolyte transport. These cells are functionally mature and play a significant role in hormone-regulated bile secretion, especially in response to secretin, which triggers cyclic AMP-dependent chloride and bicarbonate transport through CFTR and AE2 channels [10]. In contrast, small cholangiocytes, located in the smaller, peripheral branches of the intrahepatic biliary tree, have less structural complexity and lower baseline secretory function. However, they demonstrate considerable functional flexibility and can undergo compensatory proliferation in response to biliary injury or cholestatic stress [10]. In rats subjected to bile duct ligation, small cholangiocytes increase in number and can acquire secretory functions typically associated with large cholangiocytes [11]. Their ability to proliferate and regenerate contributes to bile duct repair and epithelial restoration, underscoring their emerging role in the development and recovery from cholangiopathies. Cholangiocytes exhibit functional and structural polarity, with distinct apical and basolateral membrane domains that support their specialized functions in bile production, modification, and liver homeostasis (https://abdominalkey.com/physiology-of-cholngiocytes/, last visited on 16 July 2025). The apical surface of cholangiocytes is equipped with a primary cilium, serving as a mechano- and chemosensory organelle that detects changes in bile flow and composition. This helps regulate intracellular signaling pathways essential for ductal homeostasis and bile flow maintenance [10,12].

The activation of cholangiocytes initiates molecular signaling cascades that play a role in inflammation, immune response, and tissue remodeling, defining the hallmark of cholangiopathic pathogenesis. When exposed to various injurious stimuli, cholangiocytes undergo a transformation into “reactive cholangiocytes” [13]. Once activated, cholangiocytes exhibit a unique secretory profile that significantly impacts their local microenvironment by facilitating the recruitment of immune cells and promoting the migration and activation of mesenchymal cells [14]. This includes increased expression and secretion of proinflammatory cytokines (such as interleukin (IL)-6, IL-8, tumor necrosis factor (TNF)-α), chemokines, and growth factors such as transforming growth factor-β (TGF-β), epidermal growth factor (EGF), and vascular endothelial growth factor (VEGF). These cholangiokines act through both autocrine and paracrine signaling to influence cellular processes such as proliferation, resistance to apoptosis, and cellular senescence [2].

Simultaneously, reactive cholangiocytes activate key developmental and stress-response pathways, particularly the Notch, Hedgehog, and Wnt/β-catenin cascades [2]. These signaling pathways coordinate the transcriptional programs responsible for ductular expansion, epithelial plasticity, and the initiation of fibrogenic responses [10]. The downstream effects include the recruitment of innate and adaptive immune cells, activation of hepatic stellate cells and myofibroblasts, and stimulation of angiogenesis, culminating in a ductular reaction that is clinically observed as bile duct hyperplasia [15]. This reaction is histologically characterized by the proliferation of bile ductules, periductal inflammation, matrix deposition, and often an expansion of liver progenitor cell populations. The nature of the ductular reaction, including its cellular composition and the source of proliferating cells, is highly dependent on the context and varies based on the type and severity of liver injury [15].

When left unresolved, this reactive response contributes to progressive fibrotic remodeling, leading to periportal fibrosis, ductopenia, and ultimately biliary cirrhosis, as seen in conditions such as PSC, PBC, and biliary atresia. However, it is important to emphasize the dualistic nature of cholangiocyte reactivity: under controlled circumstances, these same pathways can facilitate epithelial repair, restore bile duct integrity, and promote regeneration of the biliary tree [3]. Therefore, understanding the molecular underpinnings of cholangiocyte behavior in both injurious and reparative contexts is essential for identifying therapeutic targets and improving outcomes across the spectrum of cholangiopathies.

The portal microenvironment is comprised of a complex assembly of cell types, including biliary epithelial cells, hepatic stellate cells, portal fibroblasts, vascular endothelial cells, and diverse populations of immune cells [16]. Dynamic crosstalk among these cellular components plays a central role in initiating and perpetuating the fibrotic processes characteristic of cholangiopathies. EVs are important mediators of intercellular communication, as extensively reported in the literature, and EV-mediated signaling is increasingly being recognized in liver pathophysiology. The role played by EVs in influencing cholangiocyte activation and contributing to hepatobiliary disease is discussed in depth below.

## 3. Extracellular Vesicles in Cholangiocyte Biology and Pathology

EVs, once considered mere carriers of junk, are now seen as crucial in intercellular communication due to their role in the horizontal transfer of biomolecules such as RNA, DNA, proteins, including RNA-binding proteins, and metabolites [17,18,19]. The diverse RNA species found in EVs contribute to the heterogeneity of these small vesicles released into various body fluids, making them unique as a source of biomarkers and as a form of therapeutic support [20,21,22]. EVs come in different sizes and forms and have the ability to transport signaling molecules over long distances. Exosomes (approximately 30–150 nm) originate from multivesicular bodies (MVBs) and are released through MVB fusion with the plasma membrane, relying on ESCRT proteins (e.g., ALIX) or ceramide pathways. Microvesicles (approximately 100–1000 nm) bud directly from the plasma membrane, while apoptotic bodies (up to several micrometers) form during cellular apoptosis. EVs can be categorized based on size, cellular origin, biological function, or biogenesis. Researchers have classified EVs over time using various criteria, with specific types named according to their origin or function (e.g., ectosomes, microparticles, oncosomes, cardiosomes, vexosomes). Gould and Raposo have recommended that authors clearly define and consistently use terms, describe methods and rationale, respect terminology choices, and use “extracellular vesicle” as a general term [23]. Additionally, the International Society for Extracellular Vesicles (ISEV) suggests using operational terms for EV subtypes unless specific markers of origin are established, classifying EVs based on physical characteristics such as size or density, biochemical markers, or conditions/cell of origin (e.g., podocyte EVs, hypoxic EVs) [24]. Therefore, EVs released from biliary epithelial cells are referred to here as cholangiocyte-derived EVs. Cholangiocytes not only receive signals from EVs but are also prolific producers of EVs, participating in a complex signaling network to regulate their own phenotype and that of neighboring cells.

### 3.1. Cholangiocyte-Derived Extracellular Vesicles

An increase in the levels of biliary small extracellular vesicles (sEVs) has been observed in a cholangiopathy animal model. Although the specific mechanism underlying the increased release of EVs is not known, it is widely recognized that EVs release in cells is intricately linked to lysosomal function [25]. Several studies have shown that cells release more EVs when they are injured, especially under stress caused by excessive fat. This happens because stress and damage inside the cells disrupt normal processes, causing changes in cellular structures like lysosomes and the endoplasmic reticulum. These changes lead to more EVs being released into the environment. Overall, injury triggers cells to increase EV production through several interconnected pathways. For instance, in the context of NAFLD, lipid-induced upregulation of damage-regulated autophagy modulator (DRAM) in hepatocytes leads to the recruitment of stomatin to lysosomes. This process induces lysosomal membrane permeabilization, which impairs the degradation of multivesicular bodies by lysosomes. As a result, more MVBs fuse with the plasma membrane, thereby enhancing exosome release. This pathway highlights a critical mechanism by which lipid accumulation promotes extracellular vesicle-mediated intercellular communication during NAFLD progression [26]. We and others have demonstrated an increase in the levels of circulating EVs upon liver injuries of diverse etiologies, in preclinical models as well as in clinical samples [26,27,28,29]. Importantly, polarization, which is an important aspect of cholangiocyte biology, is faithfully reproduced in the secreted EVs. Interestingly, Davies et al. showed how polarized cholangiocytes can secrete distinct EVs from their apical and basolateral domains, in an endosomal sorting complex required for transport (ESCRT)-dependent way [27]. Normal human cholangiocytes were grown as polarized epithelial layers on Transwell inserts, enabling separate collection of culture media from their apical and basolateral sides. High resolution molecular analysis revealed that EVs secreted from the apical side were different from the basolateral ones in number and biomolecular content. For instance, the apical EVs contained a greater amount of RNA and cholesterol, and a different protein profile, compared to basal EVs [27]. This means that cholangiocytes release polarized EVs, shaped by distinct intracellular sorting mechanisms, which may impart different signaling effects in different compartments of the biliary system, as was recently described in the endothelial system [28]. Briefly, primary human aortic endothelial cells were shown to release different RNA and protein cargo in apical versus basolateral EVs, and activation by IL-1β alters both profiles. Basolateral EVs showed stronger changes linked to atherosclerosis. In silico and experimental evidence indicates that apical EVs influence monocytes, while basolateral EVs affect smooth muscle cells. Importantly, analysis of sEV-derived miRNAs from the apical and basolateral sides revealed unique biological pathways for each compartment [28]. Thus, the differential effects of apical and basolateral EVs released by cholangiocytes warrant further investigation.

### 3.2. Extracellular Vesicles Cargo Relevant to Cholangiocyte and Liver Pathology

Cholangiopathies are influenced by a complex interplay of genetic and epigenetic factors, environmental exposures, and the host-associated microbiome. This multifactorial network leads to pathological processes such as cholangiocyte senescence, immune dysregulation, and the abnormal activation of hepatic stellate cells (HSCs) and portal fibroblasts, all of which contribute to fibrogenesis and tissue remodeling. EVs play a crucial role in both initiating and amplifying these pathological interactions, making them not only central to disease progression but also promising targets for diagnostic and therapeutic approaches. Activated cholangiocytes contribute to liver fibrosis by stimulating HSCs, promoting inflammation by recruiting and activating immune cells, and contributing to cholangiocarcinogenesis through oncogenic and proliferative signaling. Cholangiocyte-derived EVs are also important mediators in paracrine communication, transferring bioactive molecules such as microRNAs, cytokines, lipids, and proteins, that modulate signaling pathways in nearby and distant cells, coordinating complex tissue responses.

### 3.3. Senescent Cholangiocyte-Associated Extracellular Vesicles

Cellular senescence is involved in the pathogenesis of numerous chronic and age-associated diseases, including liver fibrosis and hepatocellular carcinoma [29]. In PSC, an increase in the number of senescent cholangiocytes has been observed in patients’ livers compared to controls [30]. Growing research indicates that EVs released by senescent cells possess distinct properties and play a significant role in influencing the behavior of surrounding cells, similar to traditional senescence-associated secretory phenotype (SASP) factors [31]. Therefore, these EVs, known as senescence-associated EVs, can be considered an emerging component of the SASP, acting as key mediators of the cellular environment during senescence. Al Suraih et al. demonstrated that senescent cholangiocytes secrete increased quantities of EVs carrying a unique set of molecular cargo [32]. PSC-derived EVs specifically originated from senescent cells and were particularly abundant in growth factors such as amphiregulin, epidermal growth factor (EGF), and fibroblast growth factor (FGF) 7, as well as p16 and p21, and showed elevated SA-β-gal enzymatic activity. These EVs actively modulate the phenotype of neighboring cells, primarily by triggering signaling pathways involving EGF and subsequent activation of the neuroblastoma RAS (NRAS) and extracellular signal-regulated kinases 1 and 2 (ERK1/2, also known as MAPK3/1). This mechanism highlights a crucial role for cholangiocyte-derived EVs in reshaping the tissue microenvironment during cellular senescence.

### 3.4. Action on Immune System

Emerging evidence suggests that activated cholangiocytes promote the infiltration and activation of immune cells in the tissue surrounding the bile ducts by secreting chemokines and cytokines, such as monocyte chemotactic protein-1 (MCP-1) and IL-8, thereby amplifying local inflammatory responses [33]. In vitro, normal human cholangiocytes produce EVs containing 32 damage-associated molecular pattern (DAMP) molecules, including heat shock proteins, S100, growth factors, Galectin, Nucleolin, Annexin, Histone, and calreticulin, which further exacerbate macrophage activation and inflammation [34]. Among these, EV-associated S100A11 was shown to stimulate the release of pro-inflammatory cytokines, such as TNF-α, IL-1β, and IL-6, from murine macrophages.

Li et al. showed that cholangiocyte EVs robustly promoted the secretion of chemokine (C-C motif) ligand 2 (CCL-2) and IL-6 in Kupffer cells [35]. Cholangiocyte-derived EVs carry the long non-coding (lnc) RNA H19, which was shown to facilitate the development of cholestatic liver injury by disrupting bile acid homeostasis by downregulating the expression of the small heterodimer partner (SHP) in hepatocytes, a key nuclear receptor involved in the negative feedback regulation of bile acid synthesis, and promoting fibrogenesis through the activation and proliferation of HSCs [36,37]. Additionally, cholangiocyte EVs containing lncRNA H19 are promptly internalized by Kupffer cells, leading to their activation. This results in enhanced synthesis of CCL-2 and CCR-2, as well as increased expression of M1 polarization markers, including IL-6, IL12p40 (interleukin-12 subunit p40), and chemokine (C-X-C motif) ligand 10 (CXCL10) [35]. Cholangiocyte EV lncH19 could also significantly influence the chemotaxis and hepatic recruitment of bone marrow-derived macrophages (BMDMs), playing a pivotal role in modulating inflammatory cell trafficking within the liver microenvironment.

These results suggest that EVs released from activated cholangiocytes and their biomolecular cargo can potentially become a therapeutic target for cholangiopathies, for instance, targeting cholangiocyte-derived EV lncH19.

### 3.5. Fibrogenesis

Cholangiocyte-derived EVs play a critical role in regulating liver fibrogenesis by transferring bioactive molecular cargos to non-parenchymal cells, especially HSCs and liver sinusoidal endothelial cells (LSECs). In cases of cholestatic liver injury, cholangiocytes release EVs containing pro-fibrotic mediators such as long non-coding (lnc) RNAs (e.g., H19), growth factors (e.g., EGF, FGF7, and amphiregulin), and DAMPs like S100A11. These vesicles are essential for intercellular communication in cholestasis-induced liver fibrosis [16]. HSCs readily take up these EVs, influencing key signaling pathways that drive their transdifferentiation into myofibroblast-like cells, a pivotal step in the progression of liver fibrosis. Moreover, cholangiocyte-derived EVs can impact LSECs, changing their characteristics and promoting capillarization, which lead to reduced sinusoidal permeability and worsens liver fibrosis [38]. The growth factors present in EVs released by cholangiocytes can modulate fibrogenesis in different contexts (Table 1).

The overall impact of cholangiocyte-derived EVs depends on the context, influenced by the liver’s inflammatory and metabolic state, receptor distribution on target cells, and the specific signaling cascades activated. In inflammatory conditions, EVs can enhance proinflammatory signals among cholangiocytes, increasing cytokine production and cell proliferation, as demonstrated by LPS-stimulated cholangiocyte EVs [46]. For example, amphiregulin, when packaged within EVs, can be transported to target cells like HSCs or fibroblasts. Upon reaching these cells, amphiregulin binds to the EGFR, initiating downstream signaling pathways. This EGFR activation triggers pathways such as AKT, ERK1/2, and p38 MAPK, known to stimulate cell proliferation, myofibroblast differentiation, and enhanced synthesis of extracellular matrix components like collagen, key features of fibrosis [47,48]. Moreover, in immune-mediated fibrotic disorders, amphiregulin from pathogenic memory T helper 2 (Th2) cells can influence EGFR-expressing eosinophils, prompting them to release osteopontin, a potent pro-fibrotic agent. This IL-33/amphiregulin/osteopontin axis has been shown to promote fibrotic remodeling in organs such as the lung, underscoring the role of amphiregulin in fostering fibrosis through direct effects on fibroblasts and indirect modulation of immune cells [49]. However, the effects of amphiregulin can vary depending on the context. In non-fibrotic settings or during tissue repair, amphiregulin -EGFR signaling may encourage epithelial cell proliferation and tissue regeneration, potentially mitigating fibrosis by restoring tissue integrity and preventing excessive matrix deposition [50]. The anti-fibrotic functions of amphiregulin in EVs may rely on factors like the target cell condition, the presence of regulatory pathways, and the balance of other growth factors and cytokines in the surrounding environment. Understanding these mechanisms will open avenues for biomarker development and targeted therapies for cholangiopathies.

Although a comprehensive high-throughput miRNA profiling of strictly cholangiocyte-derived EVs has not been reported yet, several studies have characterized the miRNA cargo of circulating or bile-derived EVs in cholestatic conditions. In patients with PSC, circulating EVs expressing cholangiocyte markers have been profiled for miRNAs, unveiling both established and novel species [51]. In mouse models of cholestasis, such as bile duct ligation, our high-throughput sequencing has identified numerous miRNAs enriched in circulating EVs, with the liver confirmed as a primary source [52]. While these studies do not directly profile EVs exclusively from cholangiocytes, they offer valuable insights into cholestatic liver injury and suggest that findings from bile- or serum-derived EVs may help predict the potential miRNA cargo in cholangiocyte-derived EVs. Further focused research is necessary to definitively characterize the miRNA profile of cholangiocyte-specific EVs.

Overall, cholangiocyte-derived EVs orchestrate a multifaceted fibrogenic program by delivering of molecular signals that activate HSCs and other liver cell types. These EVs also impair endothelial function, and reshape the hepatic microenvironment (Figure 2).

The bidirectional exchange of EVs potentiates these pathogenic cascades. These findings highlight the potential of targeting EV-mediated intercellular communication as a therapeutic strategy in cholestatic liver diseases.

### 3.6. Extracellular Vesicle Uptake Mechanisms

Cholangiocytes internalize EVs through multiple, tightly regulated mechanisms that depend on the cellular context, origin, and their molecular cargo. These uptake pathways mainly consist of endocytic processes, such as clathrin-mediated endocytosis, non–clathrin-mediated endocytosis, and potentially receptor-mediated mechanisms. A study utilizing human cholangiocyte cell lines (H69) provided detailed insights into this process. EVs derived from the liver fluke Opisthorchis viverrini were used, demonstrating that H69 cells could uptake PKH67-labelled EVs through clathrin-dependent and non-clathrin-mediated endocytosis [53]. Treatment with chemical inhibitors like chlorpromazine or sucrose hindered clathrin-mediated endocytosis, reducing EV internalization by 80–98% compared to controls. On the other hand, bafilomycin A1, an inhibitor of non-clathrin routes involving vesicle acidification, resulted in a ~90% decrease in EV uptake. When EVs were pre-treated with proteinase K to eliminate surface proteins, uptake was reduced by ~97%, underscoring the significance of EV surface proteins interacting with cholangiocyte receptors. Additionally, pre-treating O. viverrini EVs with antibodies targeting tetraspanins (anti-TSP-2 and -3) decreased EV uptake by H69 cells by 93% and 97%, respectively. Consequently, the authors concluded that EV internalization in cholangiocytes encompasses both clathrin-dependent and independent endocytosis, facilitated by specific surface proteins on the EVs.

As described in Section 3.1, there is strong evidence that polarized cells such as cholangiocytes exhibit polarity-dependent EV release and that apical and basolateral EVs differ in composition [54]. Moreover, their signaling effects on target cells vary depending on their site of origin and the recipient cell type. While direct studies on uptake mechanisms are limited, this suggests that both the release and the functional uptake of EVs are polarity-dependent in cholangiocytes [27].

The primary cilia on cholangiocytes play a crucial role in the uptake of EVs. EVs, such as exosomes, directly interact with the primary cilia on normal rat cholangiocytes, leading to changes in ERK signaling and miR-15A expression and a reduction in proliferation [55]. Interaction between cilia and EVs is necessary for these effects, as demonstrated by the removal of cilia using chloral hydrate, which eliminates the EV-induced responses. These unique yet interconnected mechanisms of EV uptake ensure the precise delivery of signaling molecules to cholangiocytes, influencing various cellular responses like proliferation, immune modulation, and fibrosis. Understanding these pathways not only sheds light on cholangiocyte biology but also opens possibilities for therapeutic targeting interventions in cholangiopathies.

## 4. EV-Mediated Signaling in Cholangiocyte Activation

A study comparing EVs from hepatocytes and liver stem cells in *Mdr2*^−/−^ mice showed that hepatocyte-derived EVs contain lower levels of the miRNA let-7, which is important for regulating inflammation and fibrosis [56]. In these disease models, decreased let-7 levels were linked to heightened activation of pro-fibrotic and inflammatory pathways in cholangiocytes, leading to ductular reaction and biliary fibrosis. Conversely, EVs from liver stem cells, which were abundant in let-7, were able to suppress these pathways by inhibiting NF-κB and IL-13 signaling, resulting in reduced cholangiocyte proliferation and fibrosis. Therefore, hepatocyte-derived EVs have the potential to induce changes in cholangiocytes and worsen cholangiopathies by promoting inflammation and fibrosis, in contrast to stem cell-derived EVs.

In metabolic liver diseases such as NAFLD/MAFLD and NASH/MASH, metabolically-stressed hepatocytes actively secrete pro-inflammatory EVs that influence neighboring cells, including cholangiocytes. These hepatocyte-derived EVs are enriched with bioactive molecules such as tumor necrosis factor-related apoptosis-inducing ligand (TRAIL) or Vanin 1, which promote pro-inflammatory, pro-angiogenic, and pro-fibrogenic responses [57]. While their direct impact on cholangiocytes is still being elucidated, these EVs profoundly shape the hepatic microenvironment by activating macrophages and endothelial cells [57]. The resulting inflammatory milieu fosters cholangiocyte proliferation, ductular reaction, and fibrotic remodeling, which are hallmarks of disease progression in NASH [58].

Importantly, immune cell-derived EVs, such as those released by macrophages and monocytes, may play a pivotal role in modulating cholangiocyte behavior. For example, when activated by signals from stressed hepatocytes, hepatic macrophages release EVs enriched with pro-inflammatory cytokines like TNF-α and IL-6, as well as regulatory miRNAs and lipid mediators. These factors can contribute to the recruitment and polarization of monocytes and macrophages, further amplifying local immune responses. In one study, liver cells exposed to alcohol released EVs containing miR-122 which could be transferred to monocytes and inhibit the heme oxygenase-1 pathway, making monocytes more susceptible to lipopolysaccharide [59]. Additionally, it was found that macrophages exposed to environmental toxins released EVs that regulated TGF-β-related gene expression or induced α-SMA expression in fibroblasts through miRNAs or lncRNAs [60]. Although there is currently no direct evidence that EVs from immune cells specifically affect fibrogenesis in cholangiopathies, it can be envisaged that EVs can interact with toll-like receptors (TLRs) on cholangiocytes, activating signaling pathways that enhance the expression of adhesion molecules, chemokines, and other inflammatory mediators. Importantly, human cholangiocytes express several TLRs, and activation of TLR2 and TLR4 triggers downstream signaling involving MyD88 and NF-κB, leading to increased production of molecules like β-defensin-2 and various cytokines and chemokines crucial for epithelial defense and inflammation [61]. These responses may not only amplify biliary inflammation but may also contribute to cholangiocyte activation, proliferation, and ductular reaction. Platelet-derived EVs are also abundant in circulation and play a significant role in intercellular communication, including the modulation of immune and epithelial cell behavior. While direct studies on platelet-derived EVs and cholangiocytes are limited, insights obtained from related cell types indicate that platelet-derived EVs can modulate cell behavior. For example, platelet-derived EVs carry a diverse cargo of proteins, membrane receptors, and regulatory RNAs, such as miR-126, which is known for its angiogenic and reparative properties [62,63]. These EVs can be internalized by target cells, leading to changes in gene expression and cellular function. In endothelial cells, for example, platelet-derived EVs enhance angiogenesis and promote a reparative phenotype, suggesting a potential for similar effects on cholangiocytes, such as promoting proliferation, migration, or repair in response to injury [62,63]. Platelet-derived EVs may interact with immune cells, modulating inflammatory responses and potentially influencing the local microenvironment around cholangiocytes [64]. They can carry pro-inflammatory or anti-inflammatory mediators, affecting the activation state of macrophages and monocytes, which in turn can regulate cholangiocyte behavior through cytokine signaling.

In autoimmune liver diseases such as PSC and PBC, the cross-talk between immune cells and cholangiocytes, possibly through EVs, is often disrupted, resulting in a loss of immunological tolerance. This breakdown is increasingly recognized as a key factor in disease progression. This lack of interaction can facilitate antigen presentation by cholangiocytes, the attraction of immune cells through the release of chemokines and cytokines, and the maintenance of chronic inflammation. Ultimately, this process leads to progressive fibrosis of the bile ducts and surrounding liver tissue [65]. Inhibiting the NF-κB signaling pathway, for instance, can decrease the expression of these inflammatory mediators. This highlights the essential role of this pathway in coordinating immune responses and suggests potential targets for controlling inflammation in biliary and other tissues, which need further exploration [66]. Additionally, EVs can carry mitochondrial proteins or DAMPs to cholangiocytes, exacerbating stress responses and enhancing immune activation, ultimately causing tissue damage [67].

One emerging aspect of EV biology with significant implications for cholangiopathies is their prothrombotic potential, mediated primarily through the exposure of phosphatidylserine (PS) on their surface. EVs can provide a catalytic platform for the assembly of coagulation enzyme complexes, thereby promoting thrombin and fibrin generation and, hence, clot formation. This procoagulant activity has also been recently highlighted in the context of obstructive jaundice, where EVs (described as microparticles by the authors) with enhanced prothrombotic properties have been detected [68]. PS exposure on blood cells and EVs potentially contributed to enhanced shortened coagulation time and increased coagulation factor Xa, thrombin, and fibrin generation. Interestingly, increased levels of hepatic tissue necrosis, fibrin deposit and thrombophilia, reversible upon lactadherin treatment, were observed in mice subjected to bile duct ligation versus sham controls [68]. Given these findings, the prothrombotic functions of EVs warrant greater attention in future research in cholangiopathies.

Overall, hepatocyte-, immune-, and cholangiocyte-derived EVs form a complex signaling network that regulates cholangiocyte activation in cholangiopathies. Through autocrine loops, cholangiocytes drive proliferation and repair, while paracrine signals affecting macrophages, stellate cells, and endothelial cells foster inflammation and fibrosis. This layered EV communication is central to disease progression and presents novel opportunities for therapeutic intervention in cholangiopathies.

## 5. EVs in Specific Hepatobiliary Diseases and Clinical Implications

### 5.1. Primary Sclerosing Cholangitis

In PSC, a chronic immune-mediated cholangiopathy characterized by progressive bile duct inflammation and fibrosis, EVs are key mediators in connecting inflammatory and fibrogenic processes. Along with the effects of EVs from senescent cholangiocytes that promote ductular proliferation and immune cell recruitment in PSC, outer membrane vesicles (OMVs) derived from the gut microbiome also contribute to disease progression. In PSC patients with concurrent inflammatory bowel disease (IBD), gut microbiome-derived OMVs travel to the liver and trigger NLRP3 inflammasome activation in hepatic cells, exacerbating liver inflammation and fibrosis [69]. This mechanism was demonstrated using ductal organoids and *Mdr2*^−/−^ mice, highlighting the importance of vesicle-mediated intercellular communication. Additionally, analysis of circulating EVs in PSC patients has shown elevated levels of IL-13 receptor alpha 1 (IL-13Ra1), microRNA-4645-3p, and cytokeratin-19 (CK-19), which are linked to disease progression and hepatic fibrosis, suggesting their potential as non-invasive biomarkers for disease monitoring [51].

### 5.2. Cholangiocarcinoma

EVs from different cellular sources regulate the development and progression of tumors, impacting the microenvironment and tumor immunity, and ultimately affecting growth, invasion, and metastasis [70]. In CCA, EVs modulate the tumor microenvironment, promoting cholangiocyte proliferation, epithelial-to-mesenchymal transition (EMT), invasion, and migration [71]. CCA-derived EVs carry oncogenic cargo such as dihydroceramide and ceramide, which induce vascular invasion and inflammation, respectively [72]. Additionally, proteins like EGF receptor (EGFR), Mucin-1, epithelial cell adhesion molecule (EPCAM), and integrin β4 found in these EVs collectively enhance the proliferation, survival, and migratory capabilities of malignant cholangiocytes and surrounding stromal cells [73]. EVs are also involved in transforming resident fibroblasts into cancer-associated fibroblasts (CAFs), which secrete pro-inflammatory cytokines like IL-6, CXCL1, and CCL2, creating a tumorigenic niche [73]. Moreover, tumor-derived EVs suppress anti-tumor immune responses by hindering CD8^+^ T cell proliferation and activation, promoting regulatory T cell and myeloid-derived suppressor cell expansion, impairing natural killer cell function, disrupting monocyte differentiation, and inducing dysfunction and exhaustion in T cells, ultimately facilitating immune evasion and the development of drug resistance associated with cancer therapy [74,75].

Further investigation is required to fully understand how CCA-derived EVs impact tumor immunity. CCA cells release bile exosomal miR-182/183-5p, which targets hydroxyprostaglandin dehydrogenase (HPGD) in both CCA and mast cells [76]. This interaction reduces HPGD expression, leading to increased prostaglandin E2 levels and Prostaglandin E Receptor 1 (PTGER1) activation, consequently promoting CCA cell proliferation, invasion, and EMT. Additionally, miR-182/183-5p stimulate angiogenesis by upregulating VEGF-A expression, which stimulates mast cells to release VEGF-A. Mast cells have been identified as the primary cell type expressing HPGD and the primary target of bile-derived exosomal miR-182/183-5p [76]. On a clinical level, several EV-associated proteins like aminopeptidase N (APN) and pantothenate (VNN1) have been detected in the serum of CCA patients, indicating their potential as diagnostic biomarkers [77].

### 5.3. Biliary Atresia and Pediatric Disorders

Emerging research in biliary atresia and other pediatric cholangiopathies highlights the critical involvement of EV-mediated intercellular communication in early bile duct injury and disease progression. Cholangiocyte-derived EVs are enriched with lncRNAs such as H19, as well as growth factors that disrupt normal bile acid homeostasis, promote extracellular matrix deposition, and activate hepatic macrophages [35]. In experimental models of biliary atresia, EV-associated H19 RNA can promote autocrine cholangiocyte proliferation by upregulating the sphingosine-1-phosphate receptor 2 (S1PR2)–sphingosine kinase 2 (SK2) signaling pathway. The proliferative effect is primarily mediated through enhanced activation and phosphorylation of the ERK1/2 signaling cascade, driving the characteristic ductular reaction observed in neonatal biliary diseases. However, the precise mechanistic roles and cargo profiles of EVs in biliary atresia remain incompletely understood, warranting further investigation using advanced in vitro models such as cholangiocyte organoids and in vivo systems, such as biliatresone-induced developmental hepatobiliary system abnormalities, to clarify their contributions to disease onset and progression [78,79]. Incorporating organoid–immune cell co-culture systems with cholangiocyte organoids will provide a powerful platform to dissect extracellular vesicle (EV)-mediated crosstalk. This will enable detailed investigation of how EVs modulate immune responses and contribute to the pathogenesis of cholangiopathies in a physiologically relevant, multicellular context [80,81]. A deeper understanding of EV function in early life may inform novel diagnostic biomarkers and therapeutic strategies tailored for pediatric populations.

## 6. Future Directions

### 6.1. Diagnostic Potential: Extracellular Vesicles as Non-Invasive Biomarkers in Cholangiopathies

EVs have emerged as promising non-invasive biomarkers for the early detection and monitoring of cholangiopathies. They reflect the molecular changes occurring in cholangiocytes and the biliary system. Notably, EV-associated miRNAs such as miR-21 and the miR-200 family, as well as lncRNA H19, show promise for the early diagnosis of cholangiopathies such as PSC and CCA [82,83]. The specific cargo of EVs has diagnostic value that remains underexplored.

### 6.2. Therapeutic Strategies: Modulating Extracellular Vesicles and Harnessing Them for Drug Delivery

Targeting cholangiocyte-derived EVs to prevent or halt the progression of cholangiopathies is an emerging therapeutic strategy based on understanding how these EVs contribute to disease mechanisms like inflammation, biliary injury, and fibrosis. EV targeting can be achieved by inhibiting their biogenesis and release using agents like GW4869, which blocks neutral sphingomyelinase activity and reduces EV secretion [84]. Additionally, preventing the uptake of EVs by recipient cells through antagonists of surface molecules, such as heparan sulfate proteoglycans, can stop the transfer of harmful cargo [85]. Therapeutic strategies also involve engineering EVs to deliver anti-fibrotic microRNAs or drugs to counteract cholangiocyte injury [86]. These approaches, in combination with genetic or pharmacological inhibition of EV-induced signaling pathways such as TGF-β, offer promising avenues to halt the progression of cholangiopathy [87]. However, off-target effects should be taken into account. For instance, GW4869 may also influence inflammatory responses, cell viability, and other signaling pathways not directly related to EV secretion, making selective targeting of cholangiocyte-derived EVs challenging [88].

### 6.3. Technical and Biological Challenges

One of the primary biological challenges in EV research for cholangiopathies is accurately distinguishing the cell-type origin of EVs, particularly differentiating cholangiocyte-derived EVs from those released by hepatocytes. It is also important to distinguish between exosomes and microvesicles in clinical studies. This specificity is crucial for understanding disease mechanisms and developing targeted diagnostics. Rigorous functional validation of EV cargo, linking specific microRNAs, long non-coding RNAs, and proteins to pathogenic mechanisms, requires integrated in vitro and in vivo models that recapitulate cholangiopathy pathophysiology. Furthermore, continued refinement of isolation technologies, combined with robust biological validation and standardization, is essential to harness the full potential of EVs as precision biomarkers and therapeutic tools in cholangiopathies. Co-isolation of non-vesicular extracellular particles, such as lipoproteins, exomeres, and supermeres, which are present in plasma, is a significant challenge in the preparation of EVs from bile or serum [89,90]. These contaminants can affect the purity, yield, and interpretation of EV-associated biomarker studies, underscoring the need for rigorous characterization and advanced separation protocols when working with complex fluids such as bile or serum.

## 7. Conclusions

EVs have emerged as crucial players in the pathophysiology of cholangiopathies, offering transformative opportunities for both diagnostics and therapeutics. These nanoscale vesicles facilitate intricate intercellular communication within the hepatobiliary system by transferring diverse bioactive molecules that collectively regulate cholangiocyte activation, inflammation, immune interactions, and fibrogenesis. EVs originating from hepatocytes, immune cells, and cholangiocytes themselves contribute dynamically to disease processes spanning PSC, CCA, biliary atresia, and NAFLD-associated bile duct injury. The stability and molecular specificity of EV cargoes in accessible biofluids underscore their promise as non-invasive biomarkers capable of early disease detection, patient stratification, and therapeutic monitoring. Parallel advances in therapeutic modulation of EV biogenesis, cargo loading, and uptake provide novel means to disrupt pathogenic signaling cascades. Furthermore, bioengineered EVs are being developed as precision delivery vehicles to target cholangiocytes and hepatic stellate cells with antifibrotic agents, nucleic acids, or gene editing constructs. Realizing these clinical applications necessitates concerted efforts to standardize EV isolation and characterization methodologies, leverage single-EV and spatial omics technologies to map microenvironmental signaling, and employ sophisticated in vitro organoid systems alongside in vivo models for functional validation. Multidisciplinary collaboration across hepatology, molecular biology, bioengineering, and computational sciences is imperative to translate EV-based innovations from bench to bedside. Thus, EV-mediated communication represents a central axis in cholangiocyte activation and hepatobiliary disease progression and a promising frontier for biomarker discovery and therapeutic innovation, poised to usher in a new era of precision medicine in cholangiopathies.

## Figures and Tables

**Figure 1 cells-14-01274-f001:**
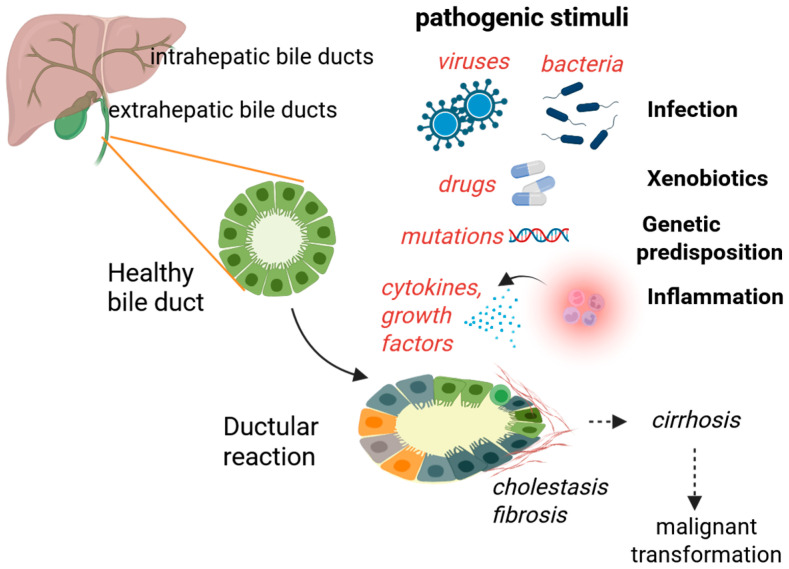
Pathogenic triggers of cholangiopathies. Bile duct injury can result from various factors, including infections by microorganisms (either from the environment or the gut), exposure to xenobiotics (like drugs or toxins), genetic predispositions (such as *MDR3* gene mutations), and immune-mediated inflammation. These insults lead to a ductular reaction characterized by cholangiocyte activation, proliferation, and, in some cases, cellular senescence. Activated cholangiocytes secrete pro-inflammatory cytokines and chemokines, interact with immune cells, and communicate with hepatic stellate cells and portal fibroblasts, which amplifies inflammatory and fibrogenic responses. The persistent ductular reaction and chronic inflammation result in the progressive deposition of the extracellular matrix, leading to periductal fibrosis. Over time, this fibrotic remodeling can progress to cirrhosis, significantly increasing the risk of malignant transformation into cholangiocarcinoma. The pathogenesis of cholangiopathies is a complex process involving genetic, environmental, and immunological factors that converge on bile duct injury, chronic inflammation, and fibrogenesis.

**Figure 2 cells-14-01274-f002:**
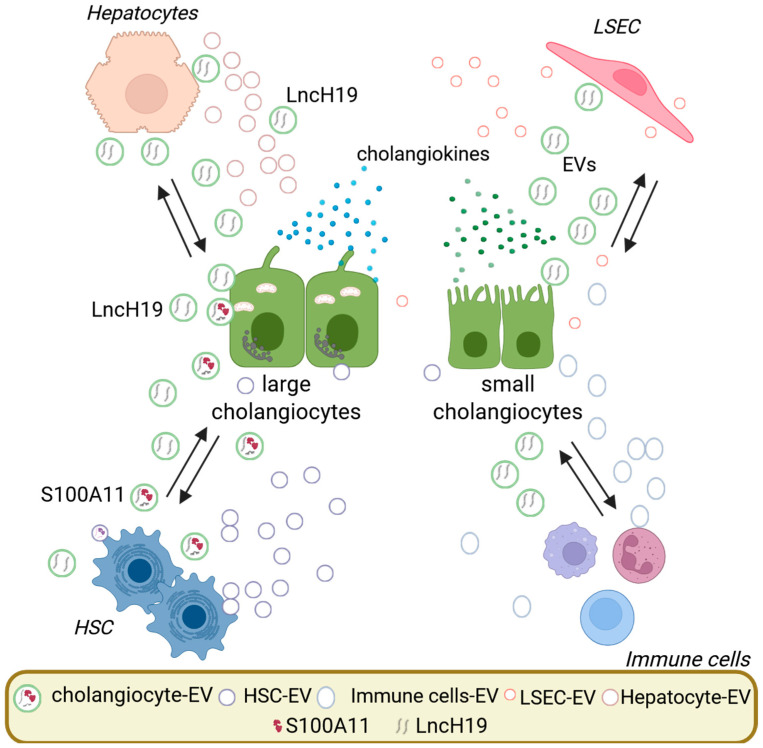
Bidirectional exchange of EVs during cholangiopathies. Activated cholangiocytes release extracellular vesicles (EVs) that carry specific molecular cargo, including the long non-coding RNA H19 (lncH19) and the damage-associated molecular pattern protein S100A11. Exosomal lncH19 from cholangiocytes is quickly taken up by hepatic stellate cells (HSCs), Kupffer cells (liver macrophages), and hepatocytes, promoting HSC activation and fibrogenesis, inducing macrophage activation and pro-inflammatory cytokine secretion, and disrupting bile acid homeostasis in hepatocytes, exacerbating liver injury and fibrosis. Similarly, EVs from cholangiocytes containing S100A11 interact with bone marrow-derived macrophages, activating them through the RAGE/NF-κB pathway and increasing periductal inflammation. Additionally, hepatocytes, HSCs, liver sinusoidal endothelial cells (LSECs), and immune cells release EVs that can impact cholangiocyte behavior, perpetuating ductular reaction, inflammation, and fibrogenesis in cholangiopathies. This intricate, bidirectional, EV-mediated communication network plays a critical role in disease progression and represents a potential target for therapeutic intervention.

**Table 1 cells-14-01274-t001:** Examples of EV-enclosed growth factors involved in liver fibrogenesis.

Growth Factor	Role in Fibrosis	Mechanism	References
Amphiregulin (EGFR ligand)	Pro-fibrotic	Upregulated in fibrotic livers in NASH. Activates EGFR on HSCs, leading to collagen production; involving AKT, ERK1/2, and p38 MAP kinases.	[39,40]
	Anti-fibrotic	Upregulated in fibrotic livers in PSC and PBC. Regulates CYP7A1 expression, serum cholesterol, and biliary acid levels through FXR.	[41]
EGF	Pro-fibrotic	EGFR activation promotes cell proliferation (myofibroblasts, HSCs), α-SMA expression, and fibrosis. Inhibiting EGFR (e.g., with EGF neutralization antibodies) reduces biliary fibrosis in rodent models.	[42,43]
	Anti-fibrotic	EGF signaling enhances intrahepatic cholangiocyte proliferation. Inhibiting EGFR (e.g., with erlotinib) dampens biliary proliferation in rodent models.	[44]
FGF7 (also known as KGF)	Generally antifibrotic/regenerative	Promotes epithelial repair and hepatocyte regeneration. Can reduce liver injury and fibrosis in some models by restoring epithelial integrity.	[45]

Abbreviations used: α-SMA: α-smooth muscle actin; CYP7A1: cholesterol 7 α-hydroxylase; EGF: epidermal growth factor; EGFR: epidermal growth factor receptor; FGF7: fibroblast growth factor 7; FXR: farnesoid X receptor; HSCs: hepatic stellate cells; KGF: keratinocyte growth factor; NASH, non-alcoholic steatohepatitis, PBC, primary biliary cholangitis; PSC, primary sclerosing cholangitis.

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
