# Peer review of "The Role of Extracellular Vesicles in Mediating Signaling in Biliary Epithelial Cell Activation and Cholangiopathies"

_cells, 2025, doi:10.3390/cells14161274_

Round 1

Reviewer 1 Report

Comments and Suggestions for Authors

This is a comprehensive literature review examining the role of extracellular vesicles (EVs) in hepatopathology, with particular emphasis on cholangiopathies. The authors effectively synthesize current literature on how EVs mediate intercellular crosstalk that contributes to dysfunctional hepatic processes—an area of intensive research interest. The review successfully identifies key knowledge gaps and provides valuable insights into future research directions, making it a solid contribution to the field.

There are some minor points that should be addressed: 
- Pg.5 Line 205-206  The finding that polarized cholangiocytes release distinct populations of apical and basolateral EVs does not provide evidence that these populations initiate different signaling effects. This should be investigated in future studies which should be specifically acknowledged.
- Pg. 6 Line 233: The study of Al Suraih et al. showed that PSC-derived EVs with particularly abundant growth factors are stemming specifically from senescent cells. This should be mentioned for the sake of clarity. 
- Pg. 6 Line 246: Explain NHC
- Pg. 8 Line 299: Based on Table 1, cholangiocyte-derived EVs exhibit both fibrogenic and antifibrogenic actions. This apparent dichotomy warrants further elaboration, particularly regarding the mechanisms that determine which pathway predominates and the factors that shift this balance toward pro- or anti-fibrotic outcomes.
- Pg. 9 Line 361: EVs derived from platelets play a substantial role as well and should be addressed.
- Pg. 10 Line 371-390: This part is somewhat speculative and should be revised since there is no particular evidence cited for EVs from immune cells affecting fibrinogenesis in cholangionpathies. The part mentioning the NF-κB signaling pathway lacks relation to EVs as well.
- Pg. 10 Line 380-381: There is a redundancy obfuscating the meaning of the sentence: dysregulated cross-talk...is often disrupted. 
- Pg 10 Line 383: This interaction...or lack of/dysregulated interaction?
- Pg. 11 Lines 414-415: The last sentence of this section is redundant
- Pg. 11 Line 437: Explain PTGER1
- Pg. 11 Line 445-446: The last sentence of this section is redundant

General comment: 
- One crucial characteristic of EVs with potential impact on cholangiopathies has been overlooked: their prothrombotic potential mediated through phosphatidylserine exposure or tissue factor activity. This aspect has been recently elucidated in obstructive jaundice and warrants inclusion in future discussions of EV-mediated cholangiopathy mechanisms. (Yu et al. BMC Gastroenterology (2025) 25:146)

Author Response

Reviewer 1:

Many thanks for reviewing our paper. We have addressed your comments/suggestions as given in the attached PDF-file.

Regards

Ralf Weiskirchen

Reviewer 2 Report

Comments and Suggestions for Authors

This is a well-written and comprehensive review that addresses an important and emerging topic: the role of extracellular vesicles in cholangiocyte activation and cholangiopathies. The manuscript is well-structured, logically progressing from basic concepts to disease-specific contexts, and effectively integrates recent literature. The figures and tables are informative, and the discussion balances mechanistic insights with translational potential.

1)     Ensure that all abbreviations (e.g., MASLD, SK2, LSECs) are defined upon their first use in the main text, even if they are included in the abbreviation list. This practice will help readers who are not specialists in hepatology.

2)     In Section 3.5, the text states that there is “no report on the miRNA profile of cholangiocyte-derived EVs,” yet it extensively discusses miR-21. Consider clarifying whether partial data exist for bile-derived or cholestasis-associated EVs and whether these findings could be extrapolated.

3)     In Section 3.6, polarity-dependent uptake is not discussed. It may be worthwhile to mention whether differences in apical versus basolateral uptake have been reported, to complement the discussion of polarized release in Section 3.1.

4)     In disease-specific sections, clarify whether the reported effects are attributable to exosomes, microvesicles, or mixed extracellular vesicle (EV) populations. This distinction is important for therapeutic targeting implications.

5)     A brief note on the possible co-isolation of non-vesicular extracellular particles (e.g., lipoproteins, exomeres) in bile or serum preparations would enhance the methodological rigor.

6)     In Section 6.2, when describing inhibitors such as GW4869, consider adding that these agents may have off-target effects and that selectively targeting cholangiocyte-derived EVs remains challenging.

7)     The figures are clear; however, the legends should be expanded to define all acronyms and briefly explain key mechanisms, making them understandable without needing to refer back to the text. Figure 2 could be improved by including molecular examples (e.g., lncH19, S100A11).

8)     Correct small typographical issues such as “in a different contexts” → “in different contexts.”

9)  Remove extra spaces before punctuation marks in several places (e.g., Section 3.1).

10) Standardize tense usage, particularly in the “Future Directions” section, where some sentences shift inconsistently between the present and future tenses.

11) In Section 5.3, when discussing cholangiocyte organoids, including a note about organoid–immune cell co-culture systems could offer forward-looking insights into dissecting extracellular vesicle (EV)-mediated crosstalk.

Author Response

Reviewer 2:

Many thanks for reviewing our paper. We have addressed your comments/suggestions as given in the attached PDF-file.

Regards

Ralf Weiskirchen
